# Simultaneous Determination of L- and D-Amino Acids in Proteins: A Sensitive Method Using Hydrolysis in Deuterated Acid and Liquid Chromatography–Tandem Mass Spectrometry Analysis

**DOI:** 10.3390/foods9030309

**Published:** 2020-03-09

**Authors:** Marianne Danielsen, Caroline Nebel, Trine Kastrup Dalsgaard

**Affiliations:** 1Department of Food Science, Aarhus University, Agro Food Park 48, 8200 Aarhus N, Denmark; 2CBIO, Centre for Circular Bioeconomy, Aarhus University, 8000 Aarhus C, Denmark; 3iFOOD, Centre for Innovative Food Research, Aarhus University, 8000 Aarhus C, Denmark

**Keywords:** D-amino acid, L-amino acid, enantiomer, LC–MS/MS, protein quality, processing, racemization, hydrolysis, food, feed

## Abstract

Determination of the L- and D-amino acid composition in proteins is important for monitoring process-induced racemization, and thereby protein quality loss, in food and feed. Such analysis has so far been challenging due to the need for sample hydrolysis, which generates racemization, thereby leading to an overestimation of D-amino acids. Here, validation of an LC–MS/MS-based method for the simultaneous determination of L- and D-amino acids in complex biological matrixes, like food and feed, was performed in combination with deuterated HCl hydrolysis. This approach eliminated a racemization-induced bias in the L- and D-amino acid ratios. The LC–MS/MS method was applied for the analysis of 18 free amino acids, with a quantification limit of either 12.5 or 62 ng/mL, except for D-phenylalanine, for which quantification was impaired by background interference from the derivatization agent. For hydrolyzed samples, the composition of 10 L- and D-amino acids pairs could be determined in protein. The average relative standard deviation was 5.5% and 6.1%, depending on the type of hydrolysis tubes. The method was applied on a green protein isolate (lucerne), which contained an average of 0.3% D-amino acids. In conclusion, this method allows for an unbiased analysis of L- and D-amino acid ratios in complex protein samples, such as food and feed.

## 1. Introduction

Food and feed proteins usually contain a very low amount of D-amino acids. The content of D-amino acids in feed [1] and foods can increase due to industrial processing, including; high temperature, extrusion or acid/alkaline treatment [2,3,4]. Fermented foods, certain health foods, and in some cases adulteration of non-fermented foods, are also known to contain higher levels of D-amino acids than fresh food [2,5,6,7]. In black vinegar, more than 50% D-amino acid has been reported for certain free amino acids [7,8]. The presence of D-amino acids in proteins leads to impaired protein digestibility and amino acid bioavailability [4]. Not all D-amino acids are metabolized in humans or animals [9] and some D-amino acids are even toxic [10].

Over the last 80 years, the consumption of industrially processed foods has dramatically increased [11]. Elevated D-amino acid levels have been reported in products such as fruit juice concentrates, syrups, break-first cereals, dairy products, olives, soya protein and bacon [2,12,13]. Furthermore, climate changes have led to a search for alternative protein sources for food and feed (monogastrics), including insect protein and protein from green plants like white clover [14]. The protein quality of these new alternatives also needs to be investigated. Therefore, being able to reliably determine protein quality, including the D-amino acid content in food and feed, is important.

An increased interest in the roles of free D-amino acids in biology has resulted in the development of sensitive and high throughput analytical methods for detecting free D-amino acids. These include quantitative LC–MS methods combined with derivatization for enantiomeric separation [15,16,17]. However, analysis of the L- and D-amino acid composition in proteins remain challenging. A major point of consideration when analyzing L- and D-amino acids in complex biological samples, such as food or feed, is the need for protein hydrolysis. Alkaline or acetic hydrolysis leads to racemization, where L-amino acids are converted to D-amino acids and vice versa. This is particularly challenging as the amount of D-amino acids in these samples can be much less than 1% of the amount of L-amino acids, which means that much more L-amino acid is converted to D-amino acids than the opposite. Consequently, if not accounted for, artificially produced D-amino acids generates a bias toward the observation of more D-amino acids in hydrolyzed samples. For a recent review on the detection and quantification of D-amino acids, see Miyamoto et al. (2017) [18]. A common procedure to overcome this problem is the 0 h extrapolation method. As outlined by Miyamato [18]; this procedure also has shortcomings, as racemization occurring in the early stages of hydrolysis before peptide bonds are broken are not taken into account. An alternative approach is the hydrogen–deuterium exchange method [19], where proteins are hydrolyzed in deuterium chloride (DCl). If an amino acid undergoes racemization, the hydrogen on the alpha carbon becomes deuterated. Consequently, amino acids or peptides undergoing racemization increase by +1Da in mass and will therefore not be included in the analysis. 

We have developed and validated a sensitive LC–MS/MS method for the analysis of 18 free proteogenic amino acids. Combined with hydrolysis using DCl, the method allowed for simulations determination of 10 L- and D-amino acids pairs in protein. In this method, hydrogen–deuterium exchange during hydrolysis results in a +1 Da in mass increase in amino acids or peptides undergoing racemization [19,20]. This means that the unbiased relative ratio between non-deuterated L- and D-amino acid can be determined in complex biological samples. We show that this method can be applied for both small volumes of liquid and for solid samples, such as food and feed. The method was used to analyze L- and D-amino acids in a protein concentrate from lucerne; a plant protein alternative to animal protein, which has a lower climate impact.

## 2. Materials and Methods 

### 2.1. Chemicals, Reagents and Materials

L- and D-amino acid standards were obtained from Sigma-Aldrich (Darmstadt, Germany). Internal standards were purchased as a “cell-free” amino acid mix of 20 stable isotope-labeled amino acids (Cambridge Isotope Laboratories Inc., Andover, MA, USA). The chiral derivatization agent (S)-N-(4-nitrophenoxycarbonyl) phenylalanine methoxyethyl ester (S-NIFE) was obtained from Santa Cruz Biotechnology (Dallas, TX, USA). All solvents for LC–MS analysis were hypergrade (Merch, Darmstadt, Germany). Amino acid standard H (ThermoFisher Scientific, Waltham, MA, USA), an 18 amino acid mix, was used as QC sample. Milli-Q water was used throughout the experiments (Millipore, Darmstadt, Germany). 

### 2.2. Standard Solutions and Calibration Standards

Stock solutions of all L- and D-amino acids were prepared individually (1 mg/mL) in the following solvents; ethanol:water (50:50, *v*/*v*) (arginine, 4-hydroxyproline, valine, leucine, histidine, lysine, proline, serine, alanine, glycine, phenylalanine, isoleucine), water (aspartic acid, glutamic acid, methionine) and 70 mM NaOH (tyrosine, threonine, tryptophan). Mixed standard solutions were further diluted with ethanol:water (50:50, *v*/*v*) to obtain working solutions at 11 different concentration levels (0, 1.25, 12.5, 125, 625, 1250, 6250, 12,500, 25,000, 37,500 and 50,000 ng/mL). Internal standard (IS) mix solution was also prepared at 1 mg/mL in ethanol:water (50:50, *v*/*v*). IS was added to calibration standards, QC and analytical samples in the ratio 1:4 (IS:sample). All solutions were kept at −80 °C until use. 

### 2.3. Protein Hydrolysis 

Protein hydrolysis was performed in either glass capillary tubes (ideal for small volumes of liquid or fine powders) or vacuum hydrolysis tubes (ideal for other types of food or feed). 

#### 2.3.1. Glass Capillary Tubes

For hydrolysis in glass capillary tubes, dimensions of 150 × 2.35 mm (Hirchmann, Eberstadt, Germany) was used. After closing one end of the glass capillary tube using a gas flame, sample was added as either a powder (protein concentrate from lucerne) or a liquid. Liquids were then evaporated to dryness using a vacuum centrifuge. Then, 40 µL of hydrolysis solution consisting of DCl (20 wt % solution in D2O, Acros organics, New Jersey, USA) + 1% mecaptoethanol and 3% phenol [21] was added to the tubes. The headspace was flushed with argon for 20 s and the capillary was quickly transferred into a gas flame to close the tube to limit the amount of oxygen. 

#### 2.3.2. Vacuum Hydrolysis Tubes

For hydrolysis in vacuum hydrolysis tubes (Vacuum Hydrolysis Tube, 1 mL, Thermo Scientific, IL, USA) sample was added to the bottom of the tube and 200 µL hydrolysis liquid was added. The sample was snap-frozen in liquid nitrogen, vacuum was applied to the tube and the tube was carefully closed. 

All samples (both glass capillary and hydrolysis tubes) were hydrolyzed at 110 °C for 20 h. Heavy labeled internal amino acid standards (mix of 20 amino acids) were added after hydrolysis. After hydrolysis, samples were centrifuged and the supernatant was evaporated to dryness and re-dissolved in either 40 µL or 200 µL ethanol:water (50:50, *v*/*v*), respectively. 

### 2.4. Sample Derivatization 

Samples were derivatized by mixing 10 µL of sample with 7 µL 0.15M sodium tetraborate and 10 µL 2.5 mg/mL (S)-NIFE in acetonitrile. The mixture was vortexed and incubated for 20 min. At this step, the reaction solution was expected to appear yellow (pH approx. 8). Undiluted hydrolyzed samples may be too acidic. In this case, more sodium tetraborate was added and the volume of H_2_O used for quenching was reduced, respectively. The reaction was quenched by adding 2 µL 4M HCl and 71 uL H_2_O. This procedure was adapted from [15,16].

### 2.5. LC–MS/MS Instrumentation and Optimization

LC–MS/MS analysis was performed on a triple quadrupole tandem mass spectrometer (6460 TripleQuad LC/MS, Agilent Technologies, Santa Clara, CA, USA) coupled to a 1290 Infinity LC system (Agilent Technologies, Santa Clara, CA, USA). Chromatographic separation was carried out on a Luna Omega C18 column (100 × 2.1 mm, 1.6 µm, 100 Å) (Phenomenex, Torrance, CA, USA) at 40 °C. Mobile phases were (A) 5% acetic acid in water and (B) 10% methanol in acetonitrile [15,16]. The LC gradient was t(min)/B (%); 0/5, 25/50, 27/98, 29/98, 29.1/5 and 40/5 operated at a flow rate of 0.25 mL/min. The injection volume was 2 µL.

Triple quadrupole MS conditions for analyzing S-NIFE derivatized L- and D- amino acids and the stable isotope-labeled IS were optimized in positive ion mode using multiple reaction monitoring (MRM). First, source parameter settings were optimized by injecting 3 amino acid standards: arginine, serine and phenylalanine. Optimization was achieved by injecting standards at multiple source parameter increments. The following source parameters were used: gas temperature, 350 °C; gas flow, 8 L/min; nebulizer, 15 psi; sheath gas temperature, 350 °C; sheath gas flow, 11 L/min; capillary voltage (positive mode) 3000 V; nozzle voltage (positive mode), 1500 V.

Then, collision voltage (CV) and fragmentor voltage were optimized for quantifier and qualifier ions for all amino acids and there IS using direct injection and the Agilent Optimizer software, version B.08.00, built 8.0.8023.0. MRM transitions were modified from Visser et al. [15] using different stable isotope-labeled standards and with quantifier and qualifier ions optimized for our instrument. The MRM transitions, parameter settings and retention times are listed in Table 1.

### 2.6. Method Validation

The method was validated in terms of linearity, lower and upper limit of quantification (LLOQ and ULOQ), stability, carry-over and with-in and between-run precision. These parameters were validated according to Guideline on bioanalytical method validation (European Medicines Agency, EMEA/CHMP/192217/2009 rev. 1 Corr. 2). Furthermore, the matrix effect and recovery after hydrolysis was evaluated.

### 2.7. Linearity and Limit of Quantification

Calibration curves ranging from 0 to 1250 ng/mL and 1250 to 50,000 ng/mL with a total of 11 calibration points were established to cover the wide range of analysis. The linear equation and the regression coefficient was obtained from an average of 3 calibration curves (individually processed from 2 individually prepared standard curve mixtures). The limit of quantification (LOQ) was established by the criteria, that the individual calibration points on the standard curve should have an accuracy of ±20% relative to the linear equation.

### 2.8. Stability and Carry-Over

Stability of the derivatized samples was evaluated by injecting the same sample every 12 hours over a 72 h period. The samples were left in the temperature-controlled autosampler at 20 °C. 

Carry-over was investigated by running a MilliQ-water sample right after a calibration curve sample with the highest concentration (50,000 ng/mL). 

### 2.9. With-In Run Accuracy and Between-Run Accuracy

To establish the with-in run and between-run accuracy, 3 identical samples for each of 4 concentration levels were made. The concentration levels spanned the entire range of analysis, with 2 concentrations points on the lower range curve, approx. LLOQ and 3×LLOQ (100 ng/mL and 300 ng/mL) and with 2 concentration points on the higher range curve at approx. 30% and 75% of ULOQ (15,000 ng/mL and 35,000 ng/mL). Four days later, new derivatization of the samples was made. These samples were analyzed on a new calibration curve prepared that day. 

### 2.10. Recovery and Matrix Effect 

Recovery was evaluated by hydrolyzing an amino acid mix with concentrations equivalent to two calibration points (6250 ng/mL and 25,000 ng/mL) in capillary tubes without IS. After hydrolysis, IS was added and the samples were analyzed on a regular standard curve without hydrolysis. The matrix effect of hydrolyzed protein was evaluated by spike-in of 12,500 ng of standard amino acid mix into hydrolyzed α-lactalbumin (capillary tubes, 10 µg/mL α-lactalbumin, *n* = 5). 

## 3. Results

### 3.1. Optimizing LC-MS Source Parameters

In order to obtain high sensitivity, the mass spectrometry source parameters were optimized to our instrumentation using three different amino acids representing different physical properties of amino acids. This optimization resulted in a 4.5, 6.8 and 8.0 times signal increase for phenylalanine, arginine and serine, respectively. Mass spectrometry parameter settings are shown in Table 1. 

### 3.2. Method Validation

The LC-MS/MS method was validated according to the description in materials and methods, Section 2.6.

#### 3.2.1. Linearity and Limit of Quantification

Linearity and limit of quantification were determined for a low range calibration curve (0–1250 ng/mL) and a high range calibration curve (1250–50,000 ng/mL). The linear regression coefficients, linear equations as well as LLOQ and ULOQ are listed in Table 2. 

The LLOQ and ULOQ were defined as the lower and upper calibration points, which were less than ±20% off, relative to the linear equation (e.g., the calibration point 12.5 ng/mL ± 2.5 ng/mL). The ULOQ was equal to the highest calibration point (50,000 ng/mL) for all analytes, whereas the LLOQ was either 12.5 ng/mL or 62.5 ng/mL for all analytes, except phenylalanine. We observed a low baseline signal from all analytes in blank derivatized samples. In contrast, we saw a very strong background signal from phenylalanine. Consequently, we were unable to establish a linear phenylalanine calibration curve for calibration points below 12,500 ng/mL. 

#### 3.2.2. Stability and Carry-Over

The derivatized samples were tested for stability. Table 3 lists the relative standard deviation in percent (%RSD) for a 72 h stability test, with injections every 12 hours. The %RSD was well below 3.5% for all amino acids, showing that the derivatized samples are stable for a three-day period at 20 °C in the autosampler. Carry-over was hardly detectable and far below the threshold of 20% of LLOQ (data not shown).

#### 3.2.3. With-In and Between-Run Precision

The with-in run and between-run precision (%RSD) for each of the four concentrations is also listed in Table 3. The with-in run precision was ≤5 %RSD for all analytes, whereas the between-run precision was ≤7.5 %RSD for all analytes. One exception was phenylalanine, where we observed a decrease in the between-run precision for the 15,000 ng/mL calibration point (17.9 %RSD), due to the high background signal for phenylalanine (see Section 3.2.1).

### 3.3. Optimization of Sample Hydrolysis

#### 3.3.1. Matrix Effect and Recovery of Hydrolyzed Samples

In order to evaluate the matrix effect of hydrolyzed protein, we made a spike-in of 12,500 ng of each amino acid standard into a solution of a hydrolyzed pure protein. We chose α-lactalbumin, as it has an amino acid composition consisting of all the amino acids we are analyzing for, with the exception of 4-hydroxy-proline. Table 4 shows the recovery (*n* = 5) and the %RSD. In general, we see a slight overestimation (4.1%–9.1%) for all amino acids. One exception is 4-hydroxy proline (2.5% underestimation), which is not present in the matrix. The %RSD was ≤5.2%. These results indicate that there is a minor matrix effect of hydrolyzed protein.

The recovery after hydrolysis was evaluated by hydrolyzing samples from two standard point concentrations: 625 ng/mL (low-level standard curve) and 25,000 ng/mL (high-level standard curve). 

After hydrolysis, internal standard was added and the samples were analyzed on the regular standard curves. The results are also shown in Table 4. The data show that for most amino acids there is a recovery of approx. 80%–90% after protein hydrolysis. 

#### 3.3.2. Comparison of Hydrolysis in Glass Capillaries and Vacuum Hydrolysis Tubes

Finally, we compared two methods of hydrolysis, using either glass capillary tubes or vacuum hydrolysis tubes. For all method validations in this paper that involved hydrolysis, the hydrolysis was performed in glass capillary tubes. As the inner diameter of these tubes is only 2.35 mm, these tubes are ideal for the hydrolysis of small volumes of liquid. However, many solid sample types, like food and feed substances, are not easily inserted into these tubes. As an alternative to glass capillary tubes, vacuum hydrolysis tubes (1 mL, ThermoFisher Scientific, Waltham, MA, USA) was used, which are ideal for the hydrolysis of larger volumes or solid samples. Therefore, we compared L- and D-amino acid analysis of a sample, hydrolyzed in either glass capillary tubes or vacuum hydrolysis tubes. For this purpose, we hydrolyzed a protein concentrate (powder) from the plant; lucerne (alfalfa). 

In Table 5, the concentration of the individual L- and D-amino acids in protein concentrate from lucerne is shown, both from hydrolysis in glass capillary tubes and from hydrolysis in vacuum hydrolysis tubes. With a few exceptions, the %RSD for the hydrolyzed samples was lower than 7%, and all %RSD values were ≤17.6%. The average %RSD was higher for D-amino acids than L-amino acids (7.9% and 4.3%, respectively). The average %RSD was slightly lower in glass capillary tubes than in vacuum hydrolysis tubes (5.5% and 6.1%, respectively). 

The proportion of D-amino acids in lucerne was very low and did not exceed 0.62% for any amino acid. The average proportion of D-amino acids hydrolyzed in glass capillaries tubes was 0.30% D-amino acids in lucerne, relative to 0.25% D-amino acids when the hydrolysis was performed in vacuum hydrolysis tubes. 

## 4. Discussion

In the development of our method for the simultaneous determination of the L- and D- amino acid composition in complex biological samples, we modified the LC–MS methods from previous publications on the analysis of free D-amino acids in body fluids [15] and tissue [16,17]. As most proteins consist of L-amino acids with very low concentrations of D-amino acids, a sensitive method combined with a wide mass range of analysis was required. In contrast to the previously published methods, we needed a much wider range of analysis in order to cover the large dynamic range of L- and D-amino acids in natural protein samples. D-amino acids appear in the low part of the analytical range (in lucerne: 3–144 ng/mL), whereas L-amino acids appear in a high range (in lucerne: 3500–30,000 ng/nL). If a single standard curve was applied over the entire analytical range, a relatively small alteration in the highest calibration points would have a large effect in the low calibration range. A more accurate quantification was made when the standard curves are split in two. Therefore, we generated two calibration curves spanning from 0 to 1250 ng/mL and 1250 to 50,000 ng/mL, respectively. Consequently, in our case, the slopes of the low range curves are a little lower than for the high range curve of the same analyte. Utilizing these calibration curves in combination with the dilution of samples (for the analysis of L-amino acids) allowed us to quantify low levels of D-amino acids in combination with high levels of L-amino acid, using the same analytical method. For an illustration of the large dynamic range required, see Figure 1 of the analysis of pure L-histidine standard before and after hydrolysis in HCl. Here, we show that the L-histidine standard contains 0.07% D-histidine and that a standard HCl hydrolysis of the same sample generates more than 10% D-histidine, supporting previous findings on hydrolysis induced racemization [18,19].

We observed a background signal from phenylalanine, which most likely derives from the chiral derivatization agent (S-NIFE) that holds phenylalanine in the chemical structure. From personal communication with Dr. Koning [15], we learned that they experienced similar problems and observed a large batch to batch variation of phenylalanine in S-NIFE. In their paper [15], the background signal from phenylalanine was reduced by the recrystallization of S-NIFE. However, this approach was not successful in our hands. As a consequence, we were unable to establish a linear phenylalanine calibration curve for calibration points below 12,500 ng/mL. A possible solution could be to synthesize another derivatization agent, which is based on a non-proteogenic amino acid instead of phenylalanine. Furthermore, as reported by Visser et al. [15], we observed that lysine and tyrosine reacted with two molecules of S-NIFE. 

When analyzing amino acids in intact food protein, in contrast to free amino acids, protein hydrolysis is required. Chemical hydrolysis (acid or alkaline) will result in a (partial) racemization of the amino acids, where L-amino acids are converted to D-amino acids and vice versa [22]. The degree of racemization depends on hydrolysis conditions, including temperature, pressure, time and chemical composition of the hydrolysis solution, as well as the individual amino acids [2,3,22]. Therefore, analysis of the L- and D-amino acid composition of hydrolyzed samples may be biased. The fraction of D-amino acids is overestimated in samples predominantly consisting of L-amino acids, as more L-amino acids than D-amino acids will racemize. This is illustrated in Figure 1, which shows the analysis of a pure L-histidine standard before (0.07% D-histidine) and after (10.4% D-histidine) hydrolysis with HCl. To overcome this issue, hydrolysis can be performed under deuterated conditions, using DCl. If an amino acid racemizes, a hydrogen–deuterium exchange will occur on the alpha carbon of the amino acid [19,20]. Consequently, racemized amino acids will increase +1 Da in mass, and will therefore not contribute to the MRM analysis of the amino acid (for an illustration of the chemistry, see [19]). Therefore, when using this hydrogen–deuterium exchange method, the correct ratio between L- and D-amino acids (before hydrolysis) is measured. DCl hydrolysis of peptides and proteins have previously been applied in different analytical set-ups, including for the identification of D-amino acid-containing neuropeptides [23]. 

We evaluated the hydrolysis method and found a minor matrix effect of hydrolyzed protein (4.1%–9.1% overestimation). In contrast, recovery after protein hydrolysis was approx. 80%–90%. Fortunately, the underestimation of the absolute quantification is partly canceled out by the matrix effect of hydrolyzed protein. More importantly, as racemization occurs evenly among L- and D-amino acids, the ratios between the enantiomers are not biased by acid hydrolysis. In the analysis of food and feed quality, it is the ratios between L- and D-amino acids, rather than the absolute values, that are important. A high D-amino acid fraction in food or feed has a negative impact on nutritional value. 

Acid hydrolysis is well known to be destructive to some amino acids [24], observed as deamidations (asparagine and glutamine) and oxidations (tryptophan and tyrosine). Therefore, aspartic acid and glutamic acid are often measured as the sum of both the acidic and the amide form. Oxidation can be minimized by the addition of phenol and mercaptoethanol. Unfortunately, in our set-up involving both DCl hydrolysis and S-NIFE derivatization, quantification that met the validation criteria could not be obtained for these amino acids and they were excluded from the analysis of hydrolyzed samples. However, the compounds were kept in the validation prior to hydrolysis, as the method also can be applied for the analysis of free amino acids.

Finally, we compared two methods of hydrolysis, using either glass capillary tubes or vacuum hydrolysis tubes. Whereas glass capillary tubes are ideal for a small volume of liquid, vacuum hydrolysis tubes are needed for hydrolysis of larger volumes or solid samples. In the analysis of a protein concentrate (powder) from the plant; lucerne (alfalfa), we found that the proportion of D-amino acids in lucerne was very low (≤0.62% for any amino acid). The average %RSD was higher for the analysis of D-amino acids than L-amino acids (7.9% and 4.3%, respectively) indicating that the precision of the method decreases a little at very low concentrations. Serine has the lowest difference in retention time between all the analyzed L- and D-enantiomers. The low retention time in combination with a very low proportion of D-serine in lucerne meant that the two enantiomers could not be distinguished in this analysis. However, in the analysis of other material with higher D-serine levels, the two compounds were successfully distinguished (data not shown).

Industrial processing or fermentation can lead to an increase in D-amino acid levels in food and feed proteins [1,2,3,4]. Consequently, the digestibility and bioavailability of these proteins decrease. So far, the determination of the L- and D-amino acid composition in proteins has been challenging, due to the need for either alkaline or acetic hydrolysis. Sample hydrolysis generates racemization, which in general leads to an overestimation of D-amino acids. 

Here, we present a validation of an LC–MS/MS-based method for the simultaneous determination of L- and D-amino acids in complex biological matrixes, like food and feed. The use of deuterated HCl for sample hydrolysis eliminated a racemization-induced bias in the L- and D-amino acid ratios. We have adapted this method for the analysis of both small volumes of liquid and for solid samples, such as food. The method was applied to the analysis of a green protein isolate from lucerne, which is a potential alternative protein source with a low climate impact.

## 5. Conclusions

We have developed a method for the simultaneous determination of L- and D-amino acids in proteins. Previously, LC–MS/MS methods have been developed for quantification of free D-amino acids using S-NIFE derivatization. We have adapted this approach for a precise determination of the L- and D-amino acid ratios in complex protein matrixes, such as food and feed. Importantly, hydrolysis induced biases introduced during sample preparation was eliminated by hydrogen–deuterium exchange. The method was applied to the analysis of a protein extract from lucerne, which contained an average of 0.3% D-amino acids. 

## Figures and Tables

**Figure 1 foods-09-00309-f001:**
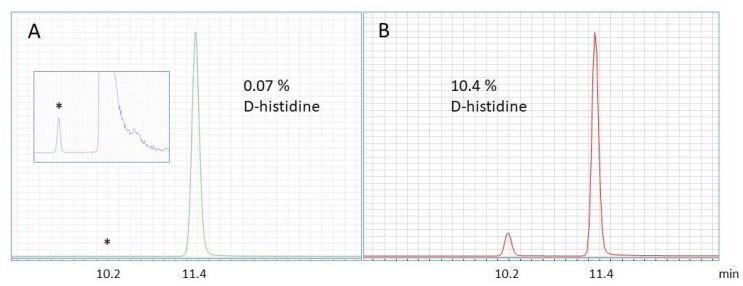
MRM spectrum of pure L-histidine standard. (**A**) L-histidine before hydrolysis contains 0.07% D-histidine. (**B**) L-histidine after hydrolysis in 6 M HCl contains 10.4% D-histidine.

**Table 1 foods-09-00309-t001:** Multiple reaction monitoring (MRM) parameters, fragmentor voltage, collision energy and retention time.

Analyte	Precursor Ion	Quantifier Ion	Qualifier Ion	Fragmentor Voltage	Collision Energy	Retention Time, L	Retention Time, D
Glycine	324.9	224.0		90	4	14.96	
			119.9	90	20		
Glycine IS	327.9	224.0		90	4		
			119.9	90	20		
Alanine	339.1	224.0		90	4	16.00	17.65
			120.0	90	20		
Alanine IS	343.1	120.0		90	4		
			224.0	90	20		
Valine	367.2	224.0		110	40	20.32	22.46
			120.0	90	8		
Valine IS	373.2	224.0		90	8		
			120.0	90	24		
Leucine	381.1	224.0		90	8	23.24	25.09
			120.0	90	28		
Leucine IS	388.1	224.0		90	8		
			120.0	90	28		
Isoleucine	381.1	224.0		90	24	22.83	24.86
			120.0	90	8		
Isoleucine IS	388.1	224.0		90	24		
			120.0	90	8		
Methionine	399.0	149.9		90	12	20.65	22.24
			120.0	90	24		
Methionine IS	405.0	224.0		90	12		
			155.9	90	12		
Phenylalanine	415.1	166.0		90	8	24.37	25.79
			120.0	90	32		
Phenylalanine IS	425.1	224.0		90	8		
			120.0	90	20		
Tryptophan	454.1	188.0		90	24	24.31	25.45
			120.0	90	28		
Tryptophan IS	467.1	224.0		90	30		
			120.0	90	30		
Proline	365.1	120.0		90	24	18.27	19.10
			114.0	90	28		
Proline IS	371.1	295.0		90	8		
			120.0	90	24		
4-hydroxy proline	380.9	131.9		90	12	13.41	
			120.0	90	24		
Serine	355.0	224.1		90	8	14.07	14.32
			120.0	90	24		
Serine IS	359.0	224.1		90	8		
			120.0	90	24		
Threonine	369.1	224.0		90	12	16.32	14.99
			120.0	90	24		
Threonine IS	374.1	224.0		90	12		
			120.0	90	12		
Tyrosine	680.2	224.0		110	20	29.91	30.05
			120.0	110	40		
Tyrosine IS	690.2	224.0		110	60		
			120.0	110	60		
Aspartic acid	383.1	133.9		90	8	14.58	15.18
			120.0	90	32		
Aspartic acid IS	388.1	224.0		90	20		
			139.0	90	8		
Glutamic acid	397.1	147.9		90	12	15.09	15.63
			120.0	90	28		
Glutamic acid IS	403.1	224.0		90	12		
			154.1	90	12		
Lysine	645.2	224.0		130	52	26.90	27.25
			120.0	130	52		
Lysine IS	653.2	224.0		130	52		
			120.0	130	52		
Arginine	424.2	201.0		130	20	12.14	11.95
			175.0	130	20		
Arginine IS	434.2	211.0		130	30		
			185.0	130	40		
Histidine	405.0	182.0		90	12	11.41	10.21
			110.0	90	36		
Histidine IS	414.0	224.0		90	28		
			120.0	90	28		

**Table 2 foods-09-00309-t002:** Linearity and limit of quantification.

Analyte	Linearity, R^2^ (0–1250 ng/mL)	Linear Equation (0–1250 ng/mL)	Linearity, R^2^ (1250–50,000 ng/mL)	Linear Equation (1250–50,000 ng/mL)	LLOQ (ng/mL)	ULOQ (ng/mL)
Glycine	0.998	y = 0.000177x − 0.000077	0.999	y = 0.000187x − 0.037257	12.5	50,000
Alanine	0.998	y = 0.000088x − 0.000107	0.999	y = 0.000095x − 0.024423	12.5	50,000
Valine	0.998	y = 0.000086x − 0.000316	0.999	y = 0.000092x − 0.024619	12.5	50,000
Leucine	0.993	y = 0.000063x + 0.000860	0.999	y = 0.000084x − 0.029608	62.5	50,000
Isoleucine	0.999	y = 0.000102x − 0.000440	0.999	y = 0.000110x − 0.03335	12.5	50,000
Methionine	0.998	y = 0.000793x − 0.005627	0.998	y = 0.000879x − 0.318433	12.5	50,000
Phenylalanine	-	-	0.989	y = 0.001942x + 0.939886	12,500	50,000
Tryptophan	0.998	y = 0.000306x − 0.001744	0.999	y = 0.000338x − 0.083823	12.5	50,000
Proline	0.998	y = 0.000322x − 0.002019	0.994	y = 0.003575x − 0.36016	12.5	50,000
4-hydroxy proline	0.995	y = 0.000128x − 0.000996	0.990	y = 0.000135x − 0.059385	12.5	50,000
Serine	0.997	y = 0.000232x + 0.00150	0.999	y = 0.000258x − 0.078782	12.5	50,000
Threonine	0.998	y = 0.000567x − 0.002145	0.999	y = 0.000597x − 0.089318	12.5	50,000
Tyrosine	0.991	y = 0.000190x − 0.00170	0.984	y = 0.000209x + 0.001621	62.5	50,000
Aspartic acid	0.995	y = 0.001266x − 0.008566	0.999	y = 0.001401x − 0.443840	62.5	50,000
Glutamic acid	0.996	y = 0.000189x − 0.001074	0.998	y = 0.000208x − 0.062358	12.5	50,000
Lysine	0.995	y = 0.000049x − 0.000379	0.999	y = 0.000055x − 0.015241	62.5	50,000
Arginine	0.995	y = 0.000149x − 0.001611	0.999	y = 0.000160x − 0.048739	62.5	50,000
Histidine	0.999	y = 0.018330x − 0.080064	0.999	y = 0.019683x − 5.175752	12.5	50,000

**Table 3 foods-09-00309-t003:** Method stability, within-run and between-run precision.

Analyte	Method Stability 0–72 h, %RSD ^1,^*	Concentration (ng/mL) ^2^	Within-Run %RSD ^2,^*	Between-Run 4 Days, %RSD ^2,^*
Glycine	1.25	100	3.02	2.36
		300	2.94	3.65
		15.000	1.38	1.22
		35.000	1.32	0.89
Alanine	0.81	100	1.02	1.56
		300	1.77	1.80
		15.000	1.26	0.87
		35.000	0.45	1.19
Valine	0.93	100	2.00	2.20
		300	3.19	3.04
		15.000	0.95	0.94
		35.000	0.23	1.20
Leucine	0.76	100	2.07	1.63
		300	2.55	2.61
		15.000	1.20	1.20
		35.000	2.54	1.48
Isoleucine	0.81	100	1.95	1.49
		300	3.10	3.12
		15.000	1.03	0.93
		35.000	0.43	1.61
Methionine	1.35	100	1.32	1.50
		300	2.22	3.06
		15.000	0.68	0.56
		35.000	0.29	1.33
Phenylalanine	-	100	-	-
		300	-	-
		15.000	3.06	17.95
		35.000	2.00	5.62
Tryptophan	1.05	100	2.26	2.68
		300	3.40	4.88
		15.000	1.11	0.78
		35.000	0.88	0.94
Proline	0.86	100	3.19	1.00
		300	0.95	2.25
		15.000	0.23	1.40
		35.000	2.07	5.44
4-hydroxy proline	1.90	100	1.84	6.97
		300	2.88	3.74
		15.000	2.55	3.91
		35.000	3.06	6.93
Serine	1.36	100	2.05	2.86
		300	1.46	4.56
		15.000	0.84	0.76
		35.000	1.07	1.91
Threonine	1.78	100	2.59	5.29
		300	3.55	2.63
		15.000	0.73	0.83
		35.000	0.19	1.53
Tyrosine	1.00	300	2.75	2.75
		400	5.00	5.00
		15.000	0.62	6.34
		35.000	1.01	7.44
Aspartic acid	3.44	100	4.07	4.10
		300	3.34	4.22
		15.000	1.20	1.40
		35.000	1.64	2.03
Glutamic acid	2.07	100	2.56	3.59
		300	2.36	2.18
		15.000	1.42	1.210
		35.000	0.61	1.15
Lysine	1.00	100	1.77	2.11
		300	1.26	1.77
		15.000	0.45	0.67
		35.000	1.84	1.49
Arginine	0.68	100	2.54	2.21
		300	3.28	2.50
		15.000	1.23	0.90
		35.000	0.86	1.75
Histidine	2.32	100	3.28	3.47
		300	1.23	2.83
		15.000	0.86	2.64
		35.000	1.02	1.95

^1^ From 0 to 72 h, 12 h intervals, *n* = 7. ^2^ Four concentrations for each analyte, *n* = 3. * Relative standard deviation in percent (%RSD).

**Table 4 foods-09-00309-t004:** Recovery after spike-in and recovery after hydrolysis.

Analyte	Recovery, % Spike-in ^1^	%RSD ^1,^*	Recovery, % (625 ng/mL) ^2^	Recovery, % (25.000 ng/mL) ^2^
Glycine	108.3	3.9	65.6	72.4
Alanine	104.7	3.2	82.8	85.4
Valine	106.4	3.3	92.3	95.0
Leucine	107.9	3.6	81.5	86.6
Isoleucine	106.7	3.5	91.8	95.8
Methionine	109.1	3.4	80.0	79.5
Proline	105.6	3.5	86.9	85.6
4-hydroxy proline	97.5	2.6	72.1	79.1
Serine	107.2	3.3	84.1	86.0
Threonine	107.2	3.9	89.1	90.5
Lysine	108.1	3.4	86.2	88.0
Arginine	104.1	4.3	88.9	85.0
Histidine	106.9	5.2	92.2	95.8

^1^ Spike-in of 12.500 ng of each amino acid into a solution of hydrolyzed α-lactalbumin, *n* = 5. ^2^ Recovery after hydrolysis of standards at two concentration. * Relative standard deviation in percent (%RSD).

**Table 5 foods-09-00309-t005:** L- and D-amino acids in protein concentrate (powder) from lucerne.

Analyte		Capillary Tubes		Vacuum Tubes		Capillary Tubes	Vacuum Tubes
		Average (ng/mg DM **)	%RSD *	Average (ng/mg DM **)	%RSD *	D-amino acid in % of total amino acid	D-amino acid in % of total amino acid
Glycine		11,119.6	3.6	10,472.7	7.0		
Alanine	L	18,478.9	4.3	18,172.8	3.7		
	D	66.4	4.6	60.8	4.8	0.36	0.33
Valine	L	22,364.0	5.3	22,624.5	2.0		
	D	39.2	3.5	32.0	2.5	0.17	0.14
Leucine	L	28,872.5	4.8	30,204.3	3.8		
	D	143.3	5.6	126.2	5.9	0.49	0.42
Isoleucine	L	16,402.4	5.8	17,320.4	1.5		
	D	4.1	17.6	3.0	7.9	0.02	0.02
Methionine	L	4396.9	4.7	3432.5	8.4		
	D	13.9	6.5	9.6	15.6	0.32	0.28
Proline	L	15,622.6	4.5	16,850.8	3.9		
	D	60.6	4.7	62.2	12.7	0.39	0.37
4-hydroxy proline	L	535.6	3.6	563.3	4.1	-	-
Serine	L+D	17,225.9	5.0	17,518.5	3.3	-	-
Threonine	L	13,895.4	5.6	14,061.0	3.6		
	D	10.0	6.8	8.5	15.2	0.07	0.06
Lysine	L	20,940.4	5.9	23,151.2	2.8		
	D	129.3	2.7	98.4	17.3	0.61	0.42
Arginine	L	22,703.1	5.2	23,643.8	4.2		
	D	72.3	4.6	61.9	4.1	0.32	0.26
Histidine	L	10,377.1	5.5	10,910.0	2.5		
	D	25.7	5.9	20.8	4.0	0.25	0.19

* Relative standard deviation in percent (%RSD). ** DM—dry matter.

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
