# Peer review of "Simultaneous Determination of L- and D-Amino Acids in Proteins: A Sensitive Method Using Hydrolysis in Deuterated Acid and Liquid Chromatography–Tandem Mass Spectrometry Analysis"

_foods, 2020, doi:10.3390/foods9030309_

Round 1

Reviewer 1 Report

Comment

The manuscript describes a method to simultaneously analyze the L- and D-amino acids using LC/tandem MS, and an application of the method for plant protein. The subject is interesting and beneficial to the related fields. The manuscript is very well prepared and written.

Minor

Table 2.

The LLOQ of phenylalanine, 12.500 or 12,500?

Paragraph 3.2.1.

Linearity was determined with two ranges (the high range and the low range with relatively less values in curve slope). Shortly, comment on the reasons for these differences in the curve slope of the linear equation.  

Reviewer 2 Report

This is an interesting article on measuring the D-amino acid found in food. Overall, the work is well performed and interesting. There have been only a few studies on this topic, especially using modern mass spectrometry approaches, and so this is timely. Only a few minor points in presentation may be worth addressing:

(1) While the authors mention that D-amino acids levels are low in foods, others have measured these in a variety of specific foods such as some health food and find high levels. For example, see the work of Hamase (10.1016/j.jchromb.2014.01.034) and others for black vinegars, and other fermented health foods.

(2) The DCL acid hydrolysis approach to digest peptides and proteins, followed by derivatization and MRM was used recently as a discovery funnel for d-amino acids in peptides (10.1021/acs.analchem.6b03658) and perhaps should be cited. They also reported that their labeled tyrosine (Tyr) was seen to have a different molecular weight than expected: two more Daltons than the label plus Tyr. Do the authors observe such unusual changes?

(3) There are amino acids that are destroyed (or modified) by acid hydrolysis: Pickering, M. V.; Newton, P. LC/GC 1990, 8, 778– 781. Did the authors observe this?

(4) The authors state they cannot measure D-phenylalanine well because of the contamination from the derivatization reagent. One could resolve this with a modified reagent based on another amino acid. Is this worth pursuing or at least mentioning?
